# Association of *OPRM1* and *OPRD1* Polymorphisms with Pain and Opioid Adverse Reactions in Colorectal Cancer

**DOI:** 10.3390/ph18020220

**Published:** 2025-02-06

**Authors:** Carolina Gutiérrez-Cáceres, Nikolas Ávila, Leslie C. Cerpa, Matías F. Martínez, Carlos E. Irarrazabal, Benjamín Torres, Olga Barajas, Nelson M. Varela, Luis A. Quiñones

**Affiliations:** 1Laboratory of Chemical Carcinogenesis and Pharmacogenetics, Department of Basic-Clinical Oncology (DOBC), Faculty of Medicine, University of Chile, Santiago 8350499, Chile; carolina.gutierrez@ciq.uchile.cl (C.G.-C.); navila@uc.cl (N.Á.); leslie.cerpa@uchile.cl (L.C.C.); benjamin.torres.1@ug.uchile.cl (B.T.); nvarela@uchile.cl (N.M.V.); 2Department of Pharmaceutical Sciences and Technology, Faculty of Chemical and Pharmaceutical Sciences, University of Chile, Santiago 8350499, Chile; matias.martinez@ciq.uchile.cl; 3Latin American Network for Implementation and Validation of Clinical Pharmacogenomics Guidelines (RELIVAF-CYTED), Santiago 8350499, Chile; 4Laboratory of Molecular and Integrative Physiology, Physiology Program, Centro de Investigación e Innovación Biomédica (CiiB), Universidad de los Andes, Santiago 7620086, Chile; cirarrazabal@uandes.cl; 5Department of Oncology, University of Chile Clinical Hospital (HCUCH), Santiago 8380453, Chile; obarajas@hcuch.cl

**Keywords:** *OPRM1*, *OPRD1*, opioid, pharmacogenetics, pharmacogenomics

## Abstract

**Background/Objectives**: Pain management in colorectal cancer is influenced by genetic variability in opioid receptor genes (*OPRM1* and *OPRD1*), potentially affecting opioid efficacy and adverse drug reactions (ADRs). This study evaluated the association of *OPRM1* (rs1799971 and rs510769) and *OPRD1* (rs2236861) polymorphisms with pain severity, opioid efficacy, and ADRs in Chilean colorectal cancer patients. **Methods**: The genotypes of *OPRM1* and *OPRD1* polymorphisms and clinical data from 69 colorectal cancer patients were analyzed. Associations between genotypes, ADRs, and pain severity (maximum Visual Analog Scale, VAS) were evaluated under inheritance models. **Results**: The *OPRM1* rs1799971 G allele was significantly associated with pain presence (*p* = 0.008), while *OPRD1* rs2236861 was linked to ADR risk (*p* = 0.042). Allelic distribution analysis revealed higher frequencies of the *OPRD1* G allele and *OPRM1* rs510769 T allele in patients with ADRs and pain, respectively. For *OPRM1* rs510769, the dominant model showed a significant association with pain severity (*p* = 0.033), while the overdominant model revealed a trend toward significance (*p* = 0.0504). Logistic regression model tests showed no significant predictive associations for the maximum VAS or ADRs under inheritance models. **Conclusions**: Genetic variations in *OPRM1* and *OPRD1* may play a role in pain perception and ADRs in colorectal cancer patients. These findings contribute to the understanding of pharmacogenomic factors in opioid therapy, emphasizing the need for further research to validate the clinical utility of these genetic markers.

## 1. Introduction

Oncological conditions are of significant basic and clinical interest, as they rank among the leading causes of morbidity and mortality worldwide [1]. In Chile and globally, cancer is one of the leading causes of death, with colorectal cancer (CRC) being one of the three most prevalent types [1,2]. In CRC patients, pain is a debilitating symptom that profoundly impacts both patients and their families [3,4]. In advanced stages, pain prevalence exceeds 70% [3]. Oncological pain is chronic and, as of 2022, is recognized as a disease in its own right, requiring long-term management [5]. Additionally, patients may experience acute pain episodes, such as incidental pain (predictable pain triggered by specific actions), breakthrough pain (sudden onset), or analgesic gap pain (end-of-dose failure) [6].

Oncological pain is classified by type—nociceptive, neuropathic, or mixed [7]—and may result from the cancer itself and/or the antineoplastic treatment employed. Another critical classification, essential for selecting the treatment and monitoring pharmacotherapeutic outcomes, is pain intensity. The most commonly used tool is the Visual Analog Scale (VAS), which ranges from 0 (no pain) to 10 (the most severe pain the patient has experienced) [8].

A 2022 systematic review reported that moderate-to-severe oncological pain affects up to 30.6% of cancer patients [9]. The pharmacotherapeutic management of such pain typically involves opioid medications combined with nonsteroidal anti-inflammatory drugs (NSAIDs) which inhibit cyclooxygenase enzymes (COX-1 and COX-2) or acetaminophen for both initiation and maintenance therapy [10]. Opioid medications are included in the WHO’s list of essential medicines for pain management and palliative care [11]. However, patients’ responses to opioids vary widely: some achieve optimal pain relief, while others experience adverse effects, poor efficacy, or develop opioid dependency syndrome [12,13].

Adverse effects associated with opioid therapy, such as nausea, constipation, respiratory depression, and opioid-induced hyperalgesia, pose significant challenges in clinical practice. Strategies to mitigate these adverse outcomes include careful dose titration, the use of adjuvant medications (e.g., antiemetics for nausea and laxatives for constipation), and rotation to different opioids when necessary. Consequently, the integration of pharmacogenetics offers a promising approach to optimize opioid therapy in terms of efficacy and safety.

Pharmacogenetics, a key tool in personalized medicine, aims to predict individual variability in drug responses by analyzing a patient’s genetic profile. This involves identifying allelic variants that may affect proteins involved in drug pharmacokinetics and pharmacodynamics [14]. Studies in animal models and twins have shown that genetic factors account for 30 to 76% of the interindividual variability in opioid requirements [15]. The United States Food and Drug Administration (FDA) currently recommends the study of pharmacogenetic biomarkers for codeine and tramadol use [16]. The Clinical Pharmacogenetics Implementation Consortium (CPIC) guidelines [17] for opioids and genes such as *CYP2D6, OPRM1*, and *COMT* provide dosage recommendations guided by the patient’s genetic profile. However, these guidelines have limited applicability and do not cover interventions or diseases not explicitly addressed.

The available clinical evidence supports the role of pharmacogenetic analyses in pain management. For *OPRM1* (*mu*-opioid receptor), the evidence provided by PharmGKB [18,19] supports its association with altered therapeutic responses to opioids. In contrast, the evidence linking polymorphisms in *OPRD1* (*delta*-opioid receptor) to opioid safety outcomes is limited. However, the pharmacological role of the delta-opioid receptor—modulating pain pathways and contributing to adverse effects such as respiratory depression and tolerance—provides a mechanistic basis for its potential relevance in opioid pharmacodynamics and safety [20].

*OPRM1* and *OPRD1* influence the pharmacodynamics of opioids, with evidence primarily focused on tramadol, codeine, buprenorphine, fentanyl, morphine, and oxycodone.

Based on this background, this study hypothesizes that polymorphisms in *OPRM1* and *OPRD1* influence the response to opioids, thereby conditioning their therapeutic effectiveness and toxicity in patients with advanced-stage colorectal cancer.

## 2. Results

### 2.1. Population

Data from 69 patients with colorectal cancer were analyzed (Table 1). Among these, 58% (40) were male, with a median age of 64 (IQR 30–90) years. Stage IV cancer was diagnosed in 39.1% (27) of the patients, with 77.8% (21) of these cases exhibiting primary metastases in the liver. Over 65% of the patients reported experiencing pain, primarily visceral and localized in the abdominal region.

Of the 69 patients, 28 (40.6%) used at least one type of opioid, with a median of two opioids per patient (IQR 1–5). A total of 62 opioids (38.3%) were observed out of 162 medications prescribed for pain management, which also included NSAIDs, paracetamol, and anticonvulsants, among others. The most frequently used opioids were tramadol (18 (29.0%)), morphine and fentanyl (16 cases each, 25.8%), buprenorphine (6(9.7%)), methadone (4 (6.5%)), and codeine (2 (3.2%)). Additionally, 10 (35.7%) of these patients required at least 1 opioid rotation, with a median of 2 opioid rotations per patient (IQR 1–4), accounting for a total of 20 rotations. Hospitalization data were available for 57 (82.6%) patients, of whom 43 (81.1%) had at least one, with a median of five hospitalizations. Among these, 26 (67.4%) patients experienced at least one pain-related hospitalization, with a median of two hospitalizations.

A total of 28 patients used opioids, with 18 of them experiencing 42 ADRs, resulting in an average prevalence of 2.3 ADRs per patient. The most frequent ADRs were nausea (n = 11, 26.2%), constipation (n = 8, 19.0%), and vomiting (n = 6, 14.3%). Causality analysis revealed that 25 ADRs (59.5%) were deemed probable, while 17 ADRs (40.5%) were classified as possible.

### 2.2. Genotypic and Allele Frequencies

The genotype distribution analysis (Table 2) revealed that for the rs1799971 variant on the *OPRM1* gene, the most frequent genotype was A/A (n = 30), followed by A/G (n = 13). Similarly, for the rs510769 variant of the same gene, the C/C genotype was predominant (n = 20) with C/T (n = 17). In the case of *OPRD1* (rs2236861), the G/G genotype was most common (n = 24) with A/G (n = 18).

The allele frequencies observed in this Chilean cohort were compared with global reference populations from the 1000 Genomes Project. The A allele frequency of *OPRM1* rs1799971 (83%) was higher than that observed in European populations (63%) and African populations (60%) and closer to frequencies in East Asian populations (83%). For *OPRM1* rs510769, the C allele frequency (67%) aligned with global averages but was slightly higher than the frequencies reported in European populations (60%). Similarly, the G allele frequency of *OPRD1* rs2236861 (76%) was significantly higher than in European (50%) and African (55%) populations. These findings underscore the importance of conducting pharmacogenomic research in diverse populations to account for genetic variability influencing drug response and safety.

### 2.3. Linkage Disequilibrium Analysis

As show in Figure 1, there is a strong linkage disequilibrium between the *OPRM1* variants, which are physically close to each other, with rs1799971 and rs510769 separated by approximately 8609 base pairs (bp) on chromosome 6. Additionally, the *OPRM1* rs1799971 and *OPRD1* rs2236861 variants, located on different chromosomes, may also exhibit functional linkage. Both results are statistically significant.

### 2.4. Association Analysis

#### 2.4.1. Association of Genotypes with ADRs and Pain

The association between genotypes of the *OPRM1* and *OPRD1* genes with ADRs and the presence of pain was analyzed. For ADRs, the *OPRM1* gene polymorphisms rs1799971 and rs510769 did not show statistically significant associations (*p* = 0.05935, and *p* = 0.3114, respectively). However, a significant association was observed with *OPRD1* rs2236861 (χ^2^ = 6.3333; *p* = 0.04214), suggesting that this variant may contribute to the occurrence of ADRs. Regarding the presence of pain, a significant association was found between the A/A genotype of *OPRM1* rs1799971 and the occurrence of pain (*p* = 0.007955), suggesting a potential influence on pain perception. This analysis included all patients, regardless of opioid use, and compared individuals reporting pain versus those without pain. In contrast, *OPRM1* rs510769 (*p* = 0.7386) and *OPRD1* rs2236861 (*p* = 0.8911) were not significantly associated with pain.

#### 2.4.2. Association of Allele Frequencies and Pain Presence and ADRs

An allelic frequency distribution of specific polymorphisms in *OPRD1* and *OPRM1* genes, comparing individuals with and without the presence of pain, was performed. A highly significant difference in allelic frequencies was observed across all analyzed polymorphisms (chi-squared test, *p* < 2^−16^). The G allele of *OPRD1* (rs2236861), A allele of OPRM1 (rs1799971), and C allele of *OPRM1* (rs510769) were more frequent in individuals experiencing pain (Table 3).

In addition, the allele frequency distributions for individuals with and without ADRs were also analyzed. Highly significant differences were observed for all analyzed polymorphisms (chi-squared test, *p* < 2^−16^). For *OPRM1* (rs1799971), the A allele was more frequent in individuals with ADRs. Similarly, for *OPRM1* (rs510769), the C allele was significantly associated with ADRs. Finally, for *OPRD1* (rs2236861), the G allele was substantially over-represented in individuals with ADRs compared to the A allele in those without ADRs (Table 4).

#### 2.4.3. Logistic Regression Models for ADRs and Pain

The association between the genotypes of the *OPRM1* and *OPRD1* genes and the occurrence of ADRs was assessed using several logistic regression models. However, no statistically significant associations were identified between the evaluated genotypes and ADR risk (Appendix A). This analysis was conducted with a subset of 28 patients who had received at least one opioid medication. Similarly, logistic regression models were used to assess the association between the genotypes of *OPRM1* and *OPRD1* and the presence of pain. Again, the analysis did not reveal statistically significant associations between the genotypes and pain presence (Appendix A).

Additionally, we have incorporated the available MME data into the ADR analyses as a covariate in logistic regression models (Appendix A). The regression analysis did not yield statistically significant results, as shown in the Appendix A, where a detailed table of the results has been provided.

#### 2.4.4. Association Between Genotypes and Effectiveness of Pain Relief

Figure 2 illustrates the association between specific genotypes of *OPRM1* (rs1799971, rs510769) and *OPRD1* (rs2236861) and the maximum Visual Analog Scale (VAS) score for pain. The Kruskal–Wallis test was performed to evaluate differences in the maximum VAS scores among different genotypes. No statistically significant differences were found for *OPRM1* rs1799971 (*p* = 0.1473), *OPRM1* rs510769 (*p* = 0.0872), or *OPRD1* rs2236861 (*p* = 0.3259).

#### 2.4.5. Association of Genotypes with Pain Severity and ADRs

Different genetic models (dominant, recessive, and overdominant) were applied for the genes *OPRM1* (rs510769) and *OPRD1* (rs2236861) to evaluate their influence on ADRs. Across all genetic models tested for both genes, no significant associations were found. Specifically, the dominant model for *OPRD1* showed a *p*-value of 0.2628, while other models showed even less notable differences in their respective comparisons. The association between pain severity and genetic variants did not reveal any significant associations between the genetic variants and maximum VAS. The closest to significance was the overdominant model of *OPRD1*, with a *p*-value of 0.1575.

The association between the *OPRM1* (rs510769) and *OPRD1* (rs2236861) genotypes and pain severity was assessed using the Kruskal-Wallis test across different genetic inheritance models, as shown in Figure 3. In the dominant model for *OPRM1* (rs510769), a significant difference was observed, with a *p*-value of 0.033, indicating that individuals carrying the CT or TT genotype might experience lower maximum pain levels compared to those with the CC genotype. In contrast, the recessive model (CC/CT vs. TT) did not show a significant association (*p* = 0.733), suggesting no difference in the maximum pain levels between TT carriers and the combined CC/CT group (Figure 3B). The overdominant model (CC/TT vs. CT) showed a trend toward significance (*p* = 0.0544), indicating a potential difference in the pain severity for individuals with the CT genotype compared to those with CC or TT genotypes (Figure 3C). On the other hand, no significant associations were observed for *OPRD1* (rs2236861) genotypes across any of the inheritance models, suggesting that this variant may not influence pain severity in the analyzed cohort.

#### 2.4.6. Association of Genotypes with Pain Severity and Number of ADRs Under Different Inheritance Models

Figure 4 illustrates the predictive models for pain and ADRs using logistic regression analyses. The odds ratios (ORs) with 95% confidence intervals are presented for the key predictors, including genetic polymorphisms and demographic factors, for each outcome. In A, the predictive model for pain shows the relationship between the *OPRM1* (rs510769, rs1799971) and *OPRD1* (rs2236861) genotypes, along with demographic factors such as sex and age at diagnosis, in predicting pain occurrence. Although none of the genotypes or demographic factors demonstrated a statistically significant association with the likelihood of pain, B depicts the predictive model for ADRs, where similar predictors were analyzed to assess their impact on ADR incidence. Again, no statistically significant predictors were identified.

## 3. Discussion

The demographic and clinical characteristics of this cohort align with the existing literature on colorectal cancer patients, particularly regarding the high prevalence of pain and opioid use. Notably, 66.7% of patients reported pain, primarily visceral and localized in the abdomen, with opioids being a common therapeutic strategy. These findings emphasize the burden of pain in advanced cancer stages and highlight the need for effective pain management tailored to individual patient profiles. The median age of 64 years is consistent with global trends in colorectal cancer epidemiology [1], and the predominance of male patients (58%) warrants further investigation into potential sex-related differences in pain perception [21].

This study explored the genetic variability in opioid receptor genes (*OPRM1* and *OPRD1*) and their association with pain and adverse drug reactions (ADRs) in colorectal cancer patients in a Latin American Mestizo sample (Chileans). While certain significant associations were identified, others underscore the complexity of genetic influence on opioid response and pain management. Our study revealed significant differences in genotypic and allelic frequencies across the polymorphisms analyzed, particularly in the *OPRM1* (rs1799971) and *OPRD1* (rs2236861) genes. The A/A genotype of *OPRM1* (rs1799971) was predominant, with a high frequency of the A allele, consistent with global [22] and Latin American studies for this variant [23].

The significant association of the G allele of *OPRD1* (rs2236861) with ADRs suggests a potential pharmacogenomic marker for opioid tolerance or side effects. This aligns with prior research suggesting a role for the *OPRD1* gene in modulating opioid efficacy and side effects. However, most research on *OPRD1* has focused on opioid addiction [24,25]. For example, Beer et al. reported significant associations between *OPRD1* variants and opioid addiction [24]. These findings fall outside the scope of our study, which focuses on ADRs and pain severity in cancer patients. This highlights the need for further research to investigate the impact of *OPRD1* variants on opioid response in the context of cancer-related pain.

The association of the A/A genotype of *OPRM1* (rs1799971) with the presence of pain reinforces the hypothesis that this variant may influence pain sensitivity and opioid response. This result appears to diverge from the findings of Yu et al., whose meta-analysis concluded that carriers of the G allele (AG + GG) required higher opioid doses for effective cancer pain management compared to AA homozygotes [26]. Several factors could contribute to this discrepancy. First, population-specific genetic variations may play a crucial role, as the meta-analysis by Yu et al. noted that the association between the G allele and increased opioid requirements was particularly significant among Asian patients. In contrast, no statistically significant correlation was observed in Caucasian populations. Our study, conducted on a Latin American cohort, may reflect genetic differences that influence the analgesic response differently from Asian or Caucasian groups.

The OPRM1 SNP rs1799971 has been widely studied for its association with the effectiveness of opioid analgesics. This variant has been shown to influence both pain sensitivity and the pharmacodynamic response to opioids, which may contribute to the heterogeneous findings reported in the literature. As highlighted in previous studies, some analyses focus on the presence or severity of pain, while others assess opioid requirements, a parameter influenced by both the level of pain and the analgesic efficacy of the opioid. This distinction is critical for interpreting the clinical impact of rs1799971, as it suggests that variability in outcomes may arise not only from differences in pain perception but also from how effectively opioids modulate pain in individuals with different genotypes. Our study assessed pain presence and severity, providing a complementary perspective to those examining opioid consumption. Future research integrating both endpoints may provide a more comprehensive understanding of this SNP’s role in opioid response.

Additionally, differences in pain assessment methodologies and clinical settings might account for these contrasting results. The Yu et al. study primarily focused on opioid dose requirements for pain management, whereas our study assessed the presence of pain itself. This variation in endpoints—opioid consumption versus pain perception—could lead to differing interpretations of the *OPRM1* polymorphism’s role.

Further analysis was conducted to evaluate the impact of different inheritance models (dominant, recessive, and overdominant) on the association of *OPRM1* and *OPRD1* polymorphisms with pain and ADR outcomes. The observed association between the *OPRM1* (rs510769) polymorphism and pain severity, particularly under the dominant inheritance model, aligns with the existing literature on the role of *OPRM1* variants in pain perception and opioid response. While specific studies on rs510769 are limited, research on other *OPRM1* polymorphisms, such as A118G (rs1799971), has demonstrated significant effects on opioid receptor function and individual variability in pain sensitivity. For instance, a study by Zhang et al. reported that the A118G variant influences opioid receptor binding and is associated with altered pain thresholds [27]. In our study, however, we did not observe all three genotype categories for the rs1799971 polymorphism, as the G/G genotype was absent. As a result, we were unable to perform inheritance model analyses for this variant. Although our study did not find significant associations for the *OPRD1* (rs2236861) polymorphism, previous research has indicated that other OPRD1 variants may play a role in pain modulation and opioid efficacy. Taken together, these findings highlight the significance of accounting for genetic variability in opioid receptor genes when assessing approaches to pain management.

Although the predictive models for pain and ADRs did not identify statistically significant predictors, the trends observed for certain genotypes (e.g., *OPRD1* rs2236861 and ADRs) highlight the complexity of pain and ADRs in clinical settings. The lack of significance may stem from the limited sample size, underscoring the need for larger studies to validate these findings. Nonetheless, the observed trends provide a rationale for incorporating pharmacogenomic testing into clinical practice to optimize pain management and minimize ADRs, especially in high-risk populations.

While numerous studies have explored the impact of genetic variability in opioid receptor genes, our research provides a unique perspective by focusing on the Chilean population, which exhibits distinct genetic traits and healthcare dynamics. To our knowledge, this is one of the few studies examining the association of *OPRM1* (rs1799971; rs510769) and *OPRD1* (rs2236861) polymorphisms with pain severity, opioid efficacy, and adverse drug reactions specifically in colorectal cancer patients from Latin America. These findings address a significant gap in pharmacogenomic research, as Latin American populations are often under-represented in global studies. Our results not only enhance the understanding of how these genetic variations influence pain perception and treatment responses but also underscore the potential for developing personalized pain management strategies tailored to this patient group. Ultimately, this work provides relevant information to clinical settings to improve the therapeutic outcomes and quality of life of colorectal cancer patients in an underserved region.

The findings underscore the importance of considering population-specific genetic variability when interpreting pharmacogenomic data. However, the limited sample size and lack of longitudinal follow-up constrain the generalizability of our results. Future studies should aim to include larger multiethnic cohorts, particularly from European populations, to validate our findings and facilitate the implementation of personalized medicine approaches.

## 4. Materials and Methods

### 4.1. Subjects

The study included patients aged 18 years or older who were diagnosed with stage III or IV colorectal cancer, receiving treatment at the National Cancer Institute or the Clinical Hospital of the University of Chile, and who provided written informed consent. The exclusion criteria comprised patients diagnosed with conditions unrelated to cancer pain or those with samples of insufficient quality for analysis.

A cohort of 69 genomic DNA samples was obtained from these patients. This study was conducted in compliance with the principles of the Declaration of Helsinki, and the protocol was approved by the Scientific Ethics Committee of the Clinical Research Center (CEIC-CEC) (register number 002146, 16 April 2024). Written informed consent was obtained from all participants prior to their inclusion in the study.

### 4.2. Sample and Clinical Data Collection

Genomic DNA was extracted from formalin-fixed, paraffin-embedded (FFPE) tissue samples or peripheral blood, depending on availability. Samples and clinical data were obtained in collaboration with the Biobank of Tissues and Fluids at the University of Chile (BTUCH). The Biobank managed all sample collection, storage, and clinical data management aspects. Clinical data were extracted from patients’ hospital medical records and organized using the REDCap data collection system. All samples were processed and stored following standardized protocols.

### 4.3. Variables

Pain severity was evaluated using the Visual Analog Scale (VAS), which ranges from 0 (no pain) to 10 (most severe pain experienced by the patient). Pain relief was assessed indirectly by reviewing the changes in pain scores after opioid administration and by documenting opioid rotation events in patients with insufficient analgesic response.

### 4.4. Rationale for SNP Selection

Potentially functional SNPs in *OPRM1* and *OPRD1* related to opioid pharmacodynamics were selected from the PharmGKB database, NCBI dbSNP [28], and the Ensembl Genome Browser [29], considering their clinical evidence levels and minor allele frequencies (MAFs) in Latin American populations. The selection criteria focused on variants with reported functional relevance, such as rs1799971 in *OPRM1*, which has been linked to changes in opioid receptor binding and pain sensitivity. This approach ensured the inclusion of clinically relevant polymorphisms to investigate genetic variability in opioid response.

### 4.5. Genomic DNA Extraction Procedure

DNA extraction from FFPE samples was performed using the Qiagen AllPrep DNA/RNA FFPE Kit (Cat. No./ID: 954734) following the manufacturer’s instructions. Briefly, freshly cut tissue sections (10–20 µm thick) with a tumor cell content of 50–80% were deparaffinized and lysed using proteinase K. After centrifugation, DNA was precipitated, while RNA remained in the supernatant. For peripheral blood samples (12–15 mL), genomic DNA was isolated using the QIAamp DNA Blood Mini Kit (Cat. No./ID: 51104) in accordance with standard protocols. The DNA quality and quantity were assessed using the 260/280 nm absorbance ratio measured with a DeNovix DS-11 spectrophotometer, complemented by agarose gel electrophoresis to ensure DNA integrity.

### 4.6. Genotyping

Genotyping was conducted using Real-Time Polymerase Chain Reaction (RT-PCR) with *TaqMan^®^* probes (Thermo Fisher Scientific, Headquartered in Waltham, MA, USA) in a Stratagene Mx3000p Real-Time PCR System (Agilent Technologies, Headquartered in Santa Clara, CA, USA).

### 4.7. Data Analysis and Statistics

Statistical analyses were conducted using RStudioTM. Descriptive analyses were performed to summarize sociodemographic and clinical variables, including sex, age, opioid usage, and other relevant factors. Continuous data were described using central tendency and dispersion measures, while categorical data were presented as frequency tables and graphs. Associations between genotypes and outcomes, such as adverse drug reactions (ADRs) and pain presence, were evaluated using the chi-squared test, while differences in maximum pain intensity (VAS scores) across genotypes were assessed using the Kruskal–Wallis test. Logistic regression models were used to estimate odds ratios (ORs) and 95% confidence intervals (CIs) to predict ADRs or pain presence based on genotypes and demographic factors.

All available variables were initially tested in univariate analyses, and the selection for the multivariate logistic regression models was based solely on statistical criteria. This approach ensured that the variables included contributed to the model’s explanatory power.

Dominant, recessive, and overdominant models were tested to explore the potential effects of different genetic inheritance patterns on pain severity and ADR risk. The overdominant model was included to account for heterozygote advantage, a phenomenon observed in certain pharmacogenomic contexts where heterozygous genotypes exhibit distinct effects compared to homozygous genotypes. This approach ensures a comprehensive evaluation of the genetic variants’ impact on clinical outcomes. Linkage disequilibrium (LD) between polymorphisms in *OPRM1* and *OPRD1* genes was also analyzed to explore potential functional relationships between variants. Linkage disequilibrium (LD) refers to the non-random association of alleles at different loci. In this study, we calculated LD coefficients using the “LD” function from the “genetics” package in R. This function computes pairwise LD measures, including D’ and r^2^, which quantify the degree of association between loci. The formula for r^2^ is:r^2^ = (D)^2^/(p_1_ × q_1_ × p_2_ × q_2_)
where D is the deviation of haplotype frequencies from expected frequencies, and p_1_, q_1_, p_2_, and q_2_ are the allele frequencies at two loci. This calculation is essential for understanding the genetic structure and for identifying regions of interest in association studies.

## Figures and Tables

**Figure 1 pharmaceuticals-18-00220-f001:**
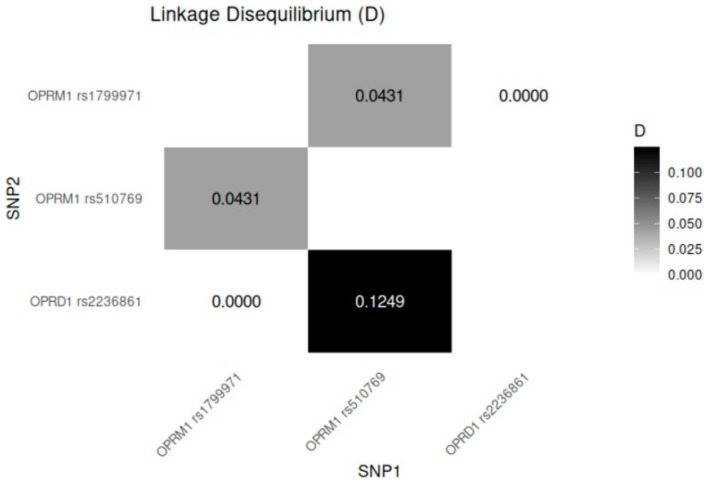
Linkage disequilibrium for variants of *OPRM1* and *OPRD1*. The values displayed in the figure represent the degree of linkage disequilibrium (D) between the analyzed SNP pairs. A value of 0 indicates no linkage, while higher values indicate stronger linkage.

**Figure 2 pharmaceuticals-18-00220-f002:**
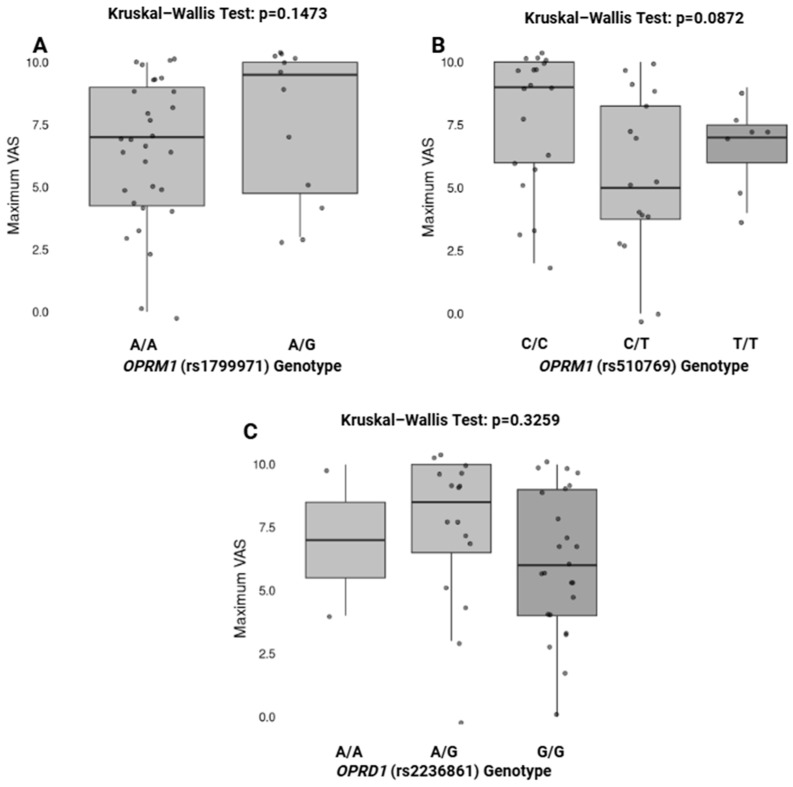
Association between genotypes and effectiveness of pain relief. *p* value < 0.05 is considered significant. Boxplots display the medians (horizontal line within each box) and the central range of data (the box). The whiskers indicate variability outside the upper and lower quartiles, and individual points represent outliers where applicable. Statistical comparisons were performed using the Kruskal-Wallis test, and the *p*-values are indicated above each plot.

**Figure 3 pharmaceuticals-18-00220-f003:**
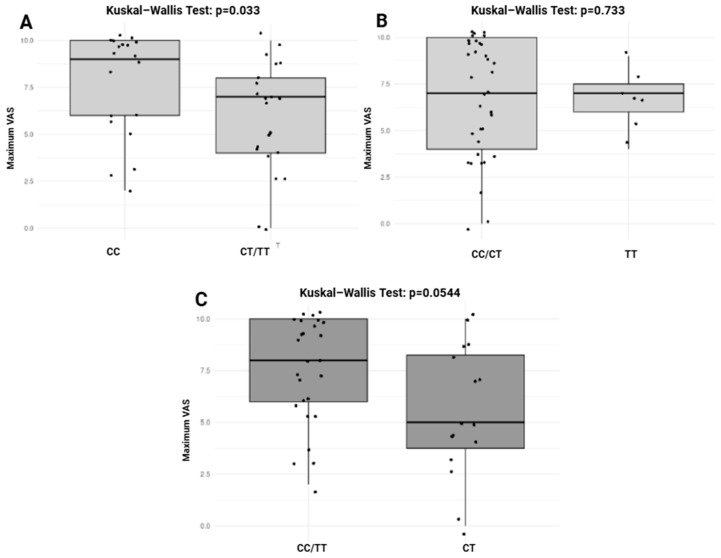
Association between *OPRM1* (rs510769) genotypes and pain severity across different genetic inheritance models. (**A**): recessive model. (**B**): dominant model. (**C**): overdominant model. *VAS*: Visual Analog Scale. *p* value < 0.05 is considered significant. Boxplots display the medians (horizontal line within each box) and the central range of data (the box). The whiskers indicate variability outside the upper and lower quartiles, and individual points represent outliers where applicable. Statistical comparisons were performed using the Kruskal-Wallis test, and the *p*-values are indicated above each plot.

**Figure 4 pharmaceuticals-18-00220-f004:**
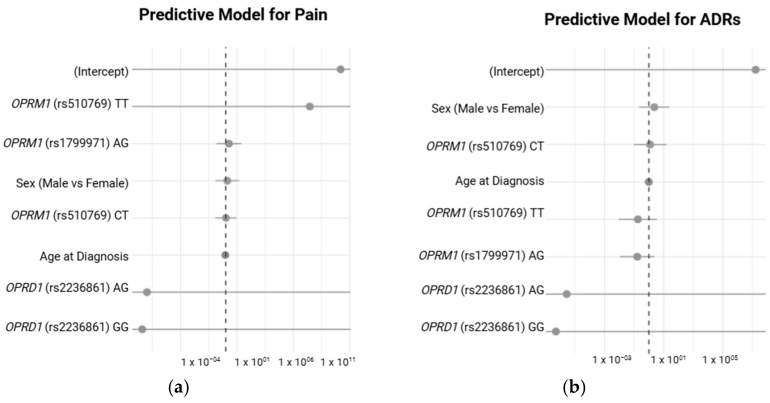
Predictive models for pain (**a**) and ADRs (**b**).

**Table 1 pharmaceuticals-18-00220-t001:** Demographic and clinical data of colorectal cancer patients (n = 69).

	n (%)
Sex	
- Female	29 (42.0%)
- Male	40 (58.0%)
Diagnostic age	
Median (range)	64 years (30–92)
Tumor localization	
- Undefined colon	23 (33.3%)
- Rectum	11 (16.0%
- Sigmoid colon	9 (13.0%)
- Other ^1^	26 (37.7%)
Cancer stage	
- III	42 (60.9%)
- IV	27 (39.1%)
Primary metastasis Location	
- Liver	21 (77.8%)
- Other ^3^	5 (18.5%)
Pain	
- Yes	46 (66.7%)
- No	5 (7.2%)
- No information	18 (26.1%)
Pain location	
- Abdomen	43 (62.3%)
- Other ^2^	23 (37.7%)
Type of pain	
- Visceral	38 (55.1%)
- Somatic	15 (21.7%)
- Neuropathic	2 (2.9%)
Opioid drug	28 (40.6%)
MME ^4^ (n = 20 *)	39.6 ± 17.2
mean ± SD	

^1^ Right colon, rectum–sigmoid, no information. ^2^ Lung, peritoneum. ^3^ Head, thorax, upper extremities, low extremities, among others. ^4^ MME: morphine milligram equivalents. * 8 patients without opioid dose information.

**Table 2 pharmaceuticals-18-00220-t002:** Genotypic and allele frequencies for *OPRM1* and *OPRD1* polymorphisms (n = 69).

Gene (Variant)	Observed Genotype Frequency	Expected Genotypic Frequency (HWE)	Allelic Frequency	*p*-Value
*OPRM1* (rs1799971)	AA: 46; AG: 23; GG: 0	AA: 47.92; AG: 19.17; GG: 1.92	A: 0.83; G: 0.17	0.097
*OPRM1* (rs510769)	CC: 32; CT: 28; TT: 9	CC: 30.67; CT: 30.67; TT: 7.67	C: 0.67; T: 0.33	0.47
*OPRD1* (rs2236861)	GG: 40; AG: 25; AA: 4	GG: 39.95; AG: 25.11; AA: 3.95	G: 0.76; A: 0.24	0.971

HWE: Hardy–Weinberg equilibrium.

**Table 3 pharmaceuticals-18-00220-t003:** Association of allele frequencies with pain presence.

Gene (Variant)	Allele	Pain Presence	Frequency	Chi-Square Test (*p*-Value)
*OPRM1* (rs1799971)	A	Pain	0.40	<2 × 10^−16^
	G	No Pain	0.00	
*OPRM1* (rs510769)	C	Pain	0.33	<2 × 10^−16^
	T	No Pain	0.10	
*OPRD1* (rs2236861)	G	Pain	0.50	<2 × 10^−16^
	A	No Pain	0.10	

**Table 4 pharmaceuticals-18-00220-t004:** Association of allele frequencies with adverse drug reactions (ADRs).

Gene (Variant)	Allele	ADR Presence	Frequency	Chi-Square Test (*p*-Value)
*OPRM1* (rs1799971)	A	ADR	0.20	<2 × 10^−16^
	G	No ADR	0.05	
*OPRM1* (rs510769)	C	ADR	0.15	<2 × 10^−16^
	T	No ADR	0.05	
*OPRD1* (rs2236861)	G	ADR	0.15	<2 × 10^−16^
	A	No ADR	0.05	

## Data Availability

Data supporting the reported results will be available upon request from the corresponding author.

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
