# Peer review of "Association of OPRM1 and OPRD1 Polymorphisms with Pain and Opioid Adverse Reactions in Colorectal Cancer"

_pharmaceuticals, 2025, doi:10.3390/ph18020220_

Round 1

Reviewer 1 Report

Comments and Suggestions for Authors

1.                  Pain is not sole cause of the death of cancer victims. I advise for including other factors such as inflammation and oxidative stress implicated in the mortality of cancer patients.

2.                  Provide reference for the following sentence.

Additionally, patients may experience acute pain episodes, such as incidental pain (predictable pain triggered by specific actions), breakthrough pain (sudden onset), or analgesic gap pain (end-of-dose failure).

3.                  Mechanism of action of NSAID should be provided.

4.                  This research is based on opoids. However, the authors did not mention any strategy for overcoming the adverse effects of opoids.

5.                  Provide reference for following sentences.

Available clinical evidence supports the role of pharmacogenetic analyses in pain management. Polymorphisms in genes such as OPRM1 (mu-opioid receptor) and OPRD1 (delta-opioid receptor) are associated with altered therapeutic responses, and polymorphisms in OPRD1 particularly linked to opioid safety outcomes. These genes influence the pharmacokinetics and pharmacodynamics of opioids, with evidence primarily focused on tramadol, codeine, buprenorphine, fentanyl, morphine, and oxycodone.

6.                  What could be the possible factors affecting the quality and quantity of genomic DNA from formalin-fixed paraffin-embedded (FFPE) tissue 318 samples or peripheral blood?

7.                  Provide the kit serial number that was used for extraction of DNA.

8.                  Agarose gel electrophoresis images of extracted DNA should be provided.

9.                  The primers sequences should be provided that were used for genotyping by Real Time PCR. Additionally, it is important to highlight the procedure of designing primers.

10.              Amplify the table ligands.

11.              The author should mention the significance of linkage of disequilibrium and formula to calculate linkage coefficient.

12.              Provide the reference.

The predominance of male patients (58%) warrants further investigation into potential sex-related differences in pain perception.

13.              A number studies have been published on the same topic. Highlight the significance of your study.

14.  This study highlights the relevance of pharmacogenomics in colorectal cancer pain management and ADR prediction in a Latin American cohort.

This sentence is not sufficient to conclude your findings. I suggest for rewriting the conclusion using a brief discussion of key findings of this study.

Reviewer 2 Report

Comments and Suggestions for Authors

Gutierrez-Caceres, et al investigated the association of polymorphisms in OPRM1 and OPRD1 with pain and adverse drug reactions (ADRs) from opioid administration in patients with colorectal cancer.  The study addresses an important clinical problem and would be of interest to others in the field.  However, I have some suggestions for improvement:

Major comments:

·       A major limitation of the study is the small sample size.  A discussion of statistical power should be added to the Methods, including the power analyses for the full cohort as would be applicable for the pain endpoint, as well as the subset of patients for the ADR analysis (N=28 patients treated with opioids).  Also, the number of patients included in each analysis should be clearly stated.  The ADR analysis is particularly underpowered and could also be confounded by the amount of opioid that a patient is receiving.  All the ADR analyses should be adjusted for the opioid dose in morphine milligram equivalents (MMEs).

·       The Methods are lacking important details.  What were the specific inclusion and exclusion criteria for the study cohort?   How were pain and pain relief assessed in the cohort?  How were variables selected for inclusion in the logistic regression models?  What was the rationale for the selection of genetic models to test (especially for including the overdominant model).  Was there any adjustment considered to account for multiple testing?

·       For the LD analysis, D’ or r should be used rather than D.

·       Please indicate whether the observed allele frequencies were consistent with Hardy-Weinberg equilibrium and how they compare with allele frequencies in the relevant reference population.

·       It is surprising that the association with allele frequencies was highly statistically significant for all SNPs, while there was not even a trend with the genotype frequencies for rs510769 and rs2236861.  Please confirm that this analysis was performed correctly.

·       Please describe the rationale for the selection of the particular SNPs to be studied in the Introduction.

·       The OPRM1 SNP rs1799971 has also been associated with the effectiveness of opioid analgesics, which may also explain some of the heterogeneous results in the literature as discussed in lines 257-268.  Some studies focused on presence of pain, while others focused on opioid requirements which is impacted both by the level of pain as well as the analgesic effectiveness of the opioid.

Minor comments:

·       In the Introduction (line 82), the statement that “polymorphisms in OPRD1 particularly linked to opioid safety outcomes” requires a citation.  Also, in the following sentence (line 83), the words “pharmacokinetics and” should be deleted since the OPRM1 and OPRD1 SNPs do not impact the pharmacokinetics of opioids.

·       Please include more details in the figure legends, including sample size when applicable.  Do the boxplots show means and standard deviations or medians and interquartile ranges?  Do the numbers in Figure 1 refer to p values or the D values?

·       Please include average opioid daily dose in MMEs for the patients taking opioids in Table 1.

·       Supplemental Figures 1 and 2 should be converted to tables and included in the main paper.

Round 2

Reviewer 2 Report

Comments and Suggestions for Authors

Thank you for the time and effort you put into addressing my comments.  Just one minor correction on line 137, the words "pharmacokinetics and" should be deleted as indicated on your response to review (top of page 7).